# Seasonal Shifts in Bacterial and Fungal Microbiomes of Leaves and Associated Leaf-Mining Larvae Reveal Persistence of Core Taxa Regardless of Diet

Hana Šigutová,[a] Martin Šigut,[a,b] Petr Pyszko,[a] Martin Kostovčík,[b] Miroslav Kolařík,[b] Pavel Drozd[a]

[a]Department of Biology and Ecology, Faculty of Science, University of Ostrava, Ostrava, Czech Republic
[b]Institute of Microbiology, Academy of Sciences of the Czech Republic, Prague, Czech Republic

**ABSTRACT** Microorganisms are key mediators of interactions between insect herbivores and their host plants. Despite a substantial interest in studying various aspects of these interactions, temporal variations in microbiomes of woody plants and their consumers remain understudied. In this study, we investigated shifts in the microbiomes of leaf-mining larvae (Insecta: Lepidoptera) and their host trees over one growing season in a deciduous temperate forest. We used 16S and ITS2 rRNA gene metabarcoding to profile the bacterial and fungal microbiomes of leaves and larvae. We found pronounced shifts in the leaf and larval microbiota composition and richness as the season progressed, and bacteria and fungi showed consistent patterns. The quantitative similarity between leaf and larval microbiota was very low for bacteria (~9%) and decreased throughout the season, whereas fungal similarity increased and was relatively high (~27%). In both leaves and larvae, seasonality, along with host taxonomy, was the most important factor shaping microbial communities. We identified frequently occurring microbial taxa with significant seasonal trends, including those more prevalent in larvae (*Streptococcus*, *Candida sake*, *Debaryomyces prosopidis*, and *Neoascochyta europaea*), more prevalent in leaves (*Erwinia*, *Seimatosporium quercinum*, *Curvibasidium cygneicollum*, *Curtobacterium*, *Ceramothyrium carniolicum*, and *Mycosphaerelloides madeirae*), and frequent in both leaves and larvae (bacterial strain P3OB-42, *Methylobacterium/Methylorubrum*, *Bacillus*, *Acinetobacter*, *Cutibacterium*, and *Botrytis cinerea*). Our results highlight the importance of considering seasonality when studying the interactions between plants, herbivorous insects, and their respective microbiomes, and illustrate a range of microbial taxa persistent in larvae, regardless of their occurrence in the diet.

**IMPORTANCE** Leaf miners are endophagous insect herbivores that feed on plant tissues and develop and live enclosed between the epidermis layers of a single leaf for their entire life cycle. Such close association is a precondition for the evolution of more intimate host-microbe relationships than those found in free-feeding herbivores. Simultaneous comparison of bacterial and fungal microbiomes of leaves and their tightly linked consumers over time represents an interesting study system that could fundamentally contribute to the ongoing debate on the microbial residence of insect gut. Furthermore, leaf miners are ideal model organisms for interpreting the ecological and evolutionary roles of microbiota in host plant specialization. In this study, the larvae harbored specific microbial communities consisting of core microbiome members. Observed patterns suggest that microbes, especially bacteria, may play more important roles in the caterpillar holobiont than generally presumed.

**KEYWORDS** bacteria, fungi, invertebrate-microbe interactions, microbial communities, microbial ecology, plant-microbe interactions

Address correspondence to Hana Šigutová, hana.sigutova@osu.cz, or Pavel Drozd, pavel.drozd@osu.cz.

The authors declare no conflict of interest.

Microorganisms have attracted considerable attention as key mediators of plant-herbivore interactions (1, 2). Leaves are inhabited by a diverse spectrum of microorganisms, predominantly bacteria and fungi (3), both on the surface (epiphytes) and in the tissues (endophytes) (4). Plant-microbe interactions range from antagonistic (plant pathogens) through neutral (commensals) to mutualistic, promoting plant fitness and growth, resilience to abiotic stressors, or resistance to pathogens (reviewed in reference 3). Leaf microbiota may also play a fundamental role in terrestrial ecosystem adaptation in the Anthropocene (5).

Likewise, insect microbiota (found mostly in the gut) may help their herbivorous hosts utilize their challenging diet by providing nutrients and interfering with and degrading plant chemical defenses (reviewed in reference 6). Although certain groups (e.g., Coleoptera and Hemiptera) have specialized structures or complex gut morphology that is able to maintain and house diverse microbial communities (7, 8), the midgut of larval Lepidoptera is a hostile environment for microbial growth due to the simple structure, high alkalinity, constantly replacing peritrophic matrix, rapid food transit, and content of host-encoded antimicrobial peptides (reviewed in reference 9). Consequently, lepidopteran microbiomes are generally thought to be species poor, reflecting mainly microbes ingested with the diet, whose functional role remains controversial (9–13).

Endophagous herbivores, such as leaf miners, feed on plant tissues and develop in a small confined space, enclosed in the two epidermis layers of a single leaf for the entire life cycle (14). Such close association is a prerequisite for the evolution of more intimate host-microbe relationships than those found in free-living herbivores (15). Indeed, the leaf-mining moth *Phyllonorycter blancardella* (Fabricius, 1781) is known to manipulate its host plant through the endosymbiotic bacterium *Wolbachia* that prevents tissue senescence through cytokinin production, thereby modifying plant phytohormonal profiles (16, 17). *Wolbachia* infections also underlie the diversification of leaf-miner moths from the Gracillariidae family (18), highlighting the evolutionary importance of microbial associations. Despite the interest in endosymbionts (16, 17, 19) or entomopathogens (20), few studies have investigated the microbiota of leaf-mining insects, and only bacteria have been targeted in previous studies (21, 22).

In forest trees, leaf microbiomes are affected by dispersal from local species pools, as well as filtering by the environment and host plant, where morphology and chemistry play important roles (23–26). Consequently, there is substantial temporal variation in microbiome richness and composition (27, 28), likely as a result of a combination of leaf senescence and microbial community succession, along with the changes in the environment and aerial microbial pool (27, 29–31).

Although seasonal changes in leaf microbiomes have been intensively studied in crops owing to their applicability in sustainable agriculture and bioenergy production (27, 32, 33), foliage of woody plants has received less attention despite its application potential in forest management. Moreover, previous studies mainly focused on evergreen trees where the effect of leaf age is limited (34–38). For deciduous temperate trees, the seasonal effect has been neglected (but see references 39 and 40). Moreover, a majority of the above-mentioned studies investigated either the bacterial (34, 36) or fungal components (35, 37, 39, 40). Similarly, seasonal shifts in microbiota of leaf-feeding larvae have not been addressed. Simultaneous comparison of microbiomes of leaves and their tightly linked consumers over time represents an interesting study system that could fundamentally contribute to the ongoing debate on the microbial residence of insect gut (see reference 11). Owing to their endophagous habit, leaf miners are ideal model organisms for interpreting the ecological and evolutionary roles of microbiota in host plant specialization.

In this study, we collected leaf-mining larvae belonging to Lepidoptera (Gracillariidae and Tischeriidae) across five host tree species over one growing season in a deciduous temperate forest. We used 16S and ITS2 rRNA gene metabarcoding to profile the bacterial and fungal microbiomes of leaves and larvae. We investigated the factors determining the richness and composition of their microbiomes, with an emphasis on seasonal

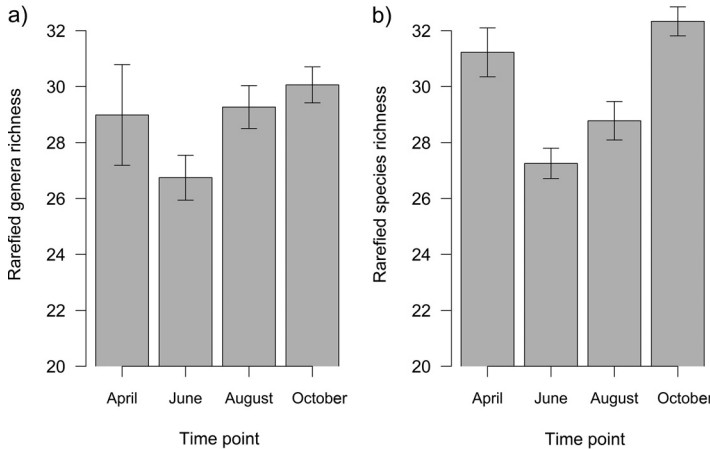

**FIG 1** Bacterial (a) and fungal (b) richness of leaf microbiota over the growing season (mean ± SE). Leaves were sampled at four time points, in April, June, August, and October, which correspond to young, mature, senescent, and old leaves, respectively. Bacterial richness did not change throughout the season (df = 361, $F$ = 1.33, $P$ = 0.249), whereas fungal richness significantly differed among time points (df = 360, $F$ = 22.75, $P$ < 0.001) as fitted by generalized linear models with gamma distribution.

variation. Specifically, we expected changes in the composition and richness of the leaf microbiome resulting from the changes in leaf quality and plant secondary metabolites during senescence (41–43), as well as microbial community succession (30). Because the diet and host physiological environment predominantly shape the microbiomes of free-feeding caterpillars (44–46), we also expected changes in larval microbiota. Moreover, we hypothesized that if the larval microbiota reflected the changes in leaf microbiota, it would suggest a transient nature with limited functionality. If larval microbiota shows a repeated occurrence of season-independent taxa, it may point out the existence of core taxa with a specific function.

## RESULTS

**Microbiota of leaves.** The bacterial community composition shifted significantly as the season progressed (13.33% of variability when comparing April versus June versus August versus October; degrees of freedom [df] = 365, $F$ = 68.92, $P$ = 0.001). Samples collected in April showed the most distinct composition from those collected at other time points (12.80% when comparing April versus the other 3 months combined; df = 365, $F$ = 53.56, $P$ = 0.001) (see Fig. S2a in the Supplemental File 1), and the composition of these samples also differed from each other (7.87% when comparing June versus August versus October; df = 274, $F$ = 23.41, $P$ = 0.001). Tree species also had a strong effect on bacterial composition (13.80%; df = 361, $F$ = 17.84, $P$ = 0.001). Other variables had markedly lower or no effect (see Table S5 in the Supplemental File 1), as the Akaike information criterion (AIC) values of the models based on these variables were higher than that of the null model.

Fungal composition also changed significantly throughout the season (5.13% when comparing April versus June versus August versus October; df = 361, $F$ = 33.38, $P$ = 0.001); however, it was more affected by tree species (29.39%; df = 362, $F$ = 47.81, $P$ = 0.001). The April samples showed the most distinct composition (3.53% when comparing April to the other 3 months combined; df = 361, $F$ = 19.0, $P$ = 0.001) (Fig. S2b, Supplemental File 1); however, the compositions of the samples collected at other time points also differed from each other (6.90% when comparing June versus August versus October; df = 270, $F$ = 18.38, $P$ = 0.001). Other variables had markedly lower or no effect (Table S5).

Bacterial richness did not change throughout the season (df = 361, $F$ = 1.33, $P$ = 0.249) (Fig. 1a) and was significantly affected only by tree species (df = 362, $F$ = 5.69, $P$ < 0.001) (Fig. S3a, Supplemental File 1). In contrast, fungal richness significantly differed among time points (df = 360, $F$ = 22.75, $P$ < 0.001) (Fig. 1b) and was also significantly affected by tree species (df = 362, $F$ = 22.75, $P$ < 0.001) (Fig. S3b, Supplemental File 1). Other significant variables with markedly lower effects are listed in Table S5.

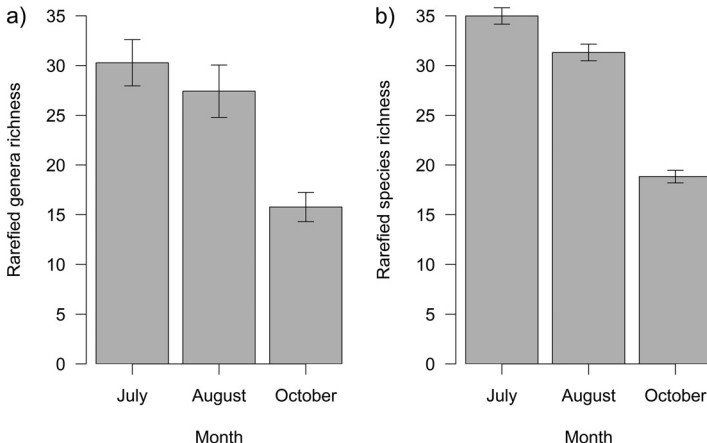

**FIG 2** Rarefied richness of larval microbiota throughout the season for bacteria (a) and fungi (b) (mean ± SE). Larvae were sampled at three time points, in June, August, and October. Bacterial richness of larval microbiota decreased throughout the season (df = 297, $F$ = 17.76, $P$ < 0.001), as well as fungal richness (df = 308, $F$ = 140.18, $P$ < 0.001), as fitted by GLM with gamma distribution and LM, respectively.

**Microbiota of larvae.** The bacterial community composition shifted significantly as the season progressed (explaining 4.66% of variability; df = 298, $F$ = 18.79, $P$ = 0.001) (Fig. S4a, Supplemental File 1), and there was a significant effect of larval species (17.64%; df = 299, F = 6.46, $P$ = 0.001) (Fig. S5a, Supplemental File 1). Other variables had markedly lower or no effect (Table S5). The fungal composition also significantly changed throughout the season (6.82%; df = 309, F = 24.39, $P$ = 0.001) (Fig. S4b, Supplemental File 1); however, there was no significant effect of the larval species (Table S5; Fig. S5b, Supplemental File 1). Other significant variables with markedly lower effects are listed in Table S5 (Supplemental File 1).

The bacterial richness of larval microbiota decreased throughout the season (df = 297, $F$ = 17.76, $P$ < 0.001) (Fig. 2a) and differed among larval species (df = 299, $F$ = 29.07, $P$ < 0.001), with the rarefied richness ranging from 7.39 to 65.18 bacterial genera in 1,000 reads per species (Fig. S6a, Supplemental File 1). Fungal richness also decreased with the season (df = 308, $F$ = 140.18, $P$ < 0.001) (Fig. 2b) and differed among larval species (df = 297, $F$ = 3.91, $P$ < 0.001), with the rarefied richness ranging from 20.73 to 36.90 fungal species in 500 reads per larval species (Fig. S6b, Supplemental File 1).

**Comparison of leaf and larval microbiota.** The quantitative similarity (Renkonen index) for bacteria was very low (~8.71 ± 0.40%) and decreased significantly as the sea-

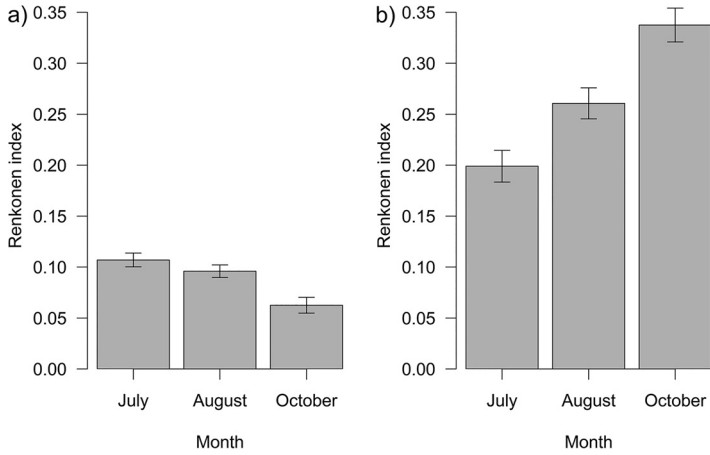

**FIG 3** Quantitative similarity between leaf and larval microbiota over the season for bacteria (a) and fungi (b) (mean ± SE). Comparisons were conducted for three time points, June, August, and October. The quantitative similarity (Renkonen index) for bacteria decreased significantly with season (df = 280, F = 18.19, $P$ < 0.001) but increased for fungi (df = 280, $\chi^2$ = 4.56, $P$ = 0.033) as fitted by GLM with binomial distribution.

son progressed (df = 280, $F$ = 18.19, $P$ < 0.001) (Fig. 3a). In addition, the index differed significantly among sampling plots (df = 278, $F$ = 2.44, $P$ = 0.033). The Renkonen index for fungi was higher than that for bacteria (~27.03 ± 0.93%) and increased significantly throughout the season (df = 280, $\chi^2$ = 4.56, $P$ = 0.033), and the same trend was observed for endophytes (~52.62 ± 1.63%; df = 280, $\chi^2$ = 5.95, $P$ = 0.015) (Fig. 3b).

The bacterial composition of leaves and larvae differed significantly (23.82% of variability; df = 585, $F$ = 182.92, $P$ = 0.001) (Fig. 4a), and the sample type (leaf or larva; corrected AIC [$AIC_c$] = 3,005.89) explained the variability better than time point ($AIC_c$ = 3,145.46; 3.37% of variability, $F$ = 20.42, $P$ = 0.001), tree species ($AIC_c$ = 3145.61; 4.34%, $F$ = 6.61, $P$ = 0.001), or sampling plot ($AIC_c$ = 3,166.19; 0.24%, $F$ = 0.71, $P$ = 0.845). The overall bacterial $\beta$-diversity of leaves and larvae differed significantly, with larvae showing higher variability than leaves (df = 585, $F$ = 227.85, $P$ < 0.001). The fungal composition of leaves and larvae also differed significantly; the amount of explained variability was more than twice as low as in the case of bacteria (10.07%; df = 585, $F$ = 65.52, $P$ = 0.001) (Fig. 4b). However, the sample type ($AIC_c$ = 3,010.69) explained the variability better than tree species ($AIC_c$ = 3,040.92; 6.30%, $F$ = 9.78, $P$ = 0.001), time point ($AIC_c$ = 3,055.06; 3.01%, $F$ = 18.16, $P$ = 0.001), or sampling plot ($AIC_c$ = 3,070.76; 0.72%, $F$ = 2.13, $P$ = 0.002). The results of the same analysis limited to fungal endophytes were similar (13.07%, df = 585, $F$ = 87.94, $P$ = 0.001; sample type, $AIC_c$ = 2,888.63; tree, $AIC_c$ = 2,951.21, 4.28%, $F$ = 6.51, $P$ = 0.001; time point, $AIC_c$ = 2,947.82, 3.84%, $F$ = 23.39, $P$ = 0.001; plot, $AIC_c$ = 2,967.24, 0.95%, $F$ = 2.80, $P$ = 0.003). The overall fungal $\beta$-diversity of leaves and larvae differed significantly, and the fungal $\beta$-diversity of larvae was more variable than that of leaves (df = 585, $F$ = 500.44, $P$ < 0.001). The similarity in fungal compositions of leaf and larval samples increased throughout the season.

The hierarchical taxonomic composition of the leaf and larval microbiota throughout the season (along with the number of reads of respective taxa) is shown in Fig. S7 (bacteria; available at https://figshare.com/articles/figure/Supplementary_figure_Fig_S7_for _Seasonal_shifts_in_bacterial_and_fungal_microbiomes_of_leaves_and_associated_leaf-mining_larvae_reveal_persistence_of_core_taxa_regardless_of_diet/20490144) and Fig. S8 (fungi; available at https://figshare.com/articles/figure/Supplementary_figure_Fig_S8 _for_Seasonal_shifts_in_bacterial_and_fungal_microbiomes_of_leaves_and_associated_ leaf-mining_larvae_reveal_persistence_of_core_taxa_regardless_of_diet/20490486). The most frequent microbial taxa with identified seasonal trends included *Methylobacterium/ Methylorubrum*, *Cutibacterium*, and *Bacillus* (bacteria) and *Ampelomyces quisqualis*, *Ceramothyrium carniolicum*, and *Candida sake* (fungi) (Table S6; Fig. 5). Neutral models revealed that in larvae, the bacterial assemblages were generally less stochastic than the fungal assemblages (23.4% versus 19.9% of the taxa out of prediction, respectively) (Fig. 6). Microbial taxa with identified seasonal trends that occurred more frequently than predicted by neutral models are highlighted in Table S6 in the Supplemental File 1.

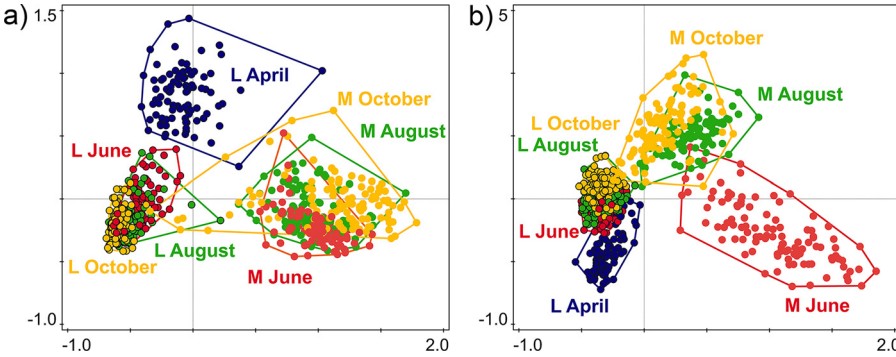

**FIG 4** Principal-coordinate analysis (PCoA) based on CCA with a combination of sample type (L, leaf; M, leaf miner larva) and time point (April, June, August, and October) as explanatory variables for bacteria (a) and fungi (b), explaining 21.30% (df = 672, $F$ = 30.30, $P$ = 0.001) and 14.20% of variability (df = 672, $F$ = 18.50, $P$ = 0.001), respectively. Unlike the PERMANOVA results, the PCoA plots show data for all time points.

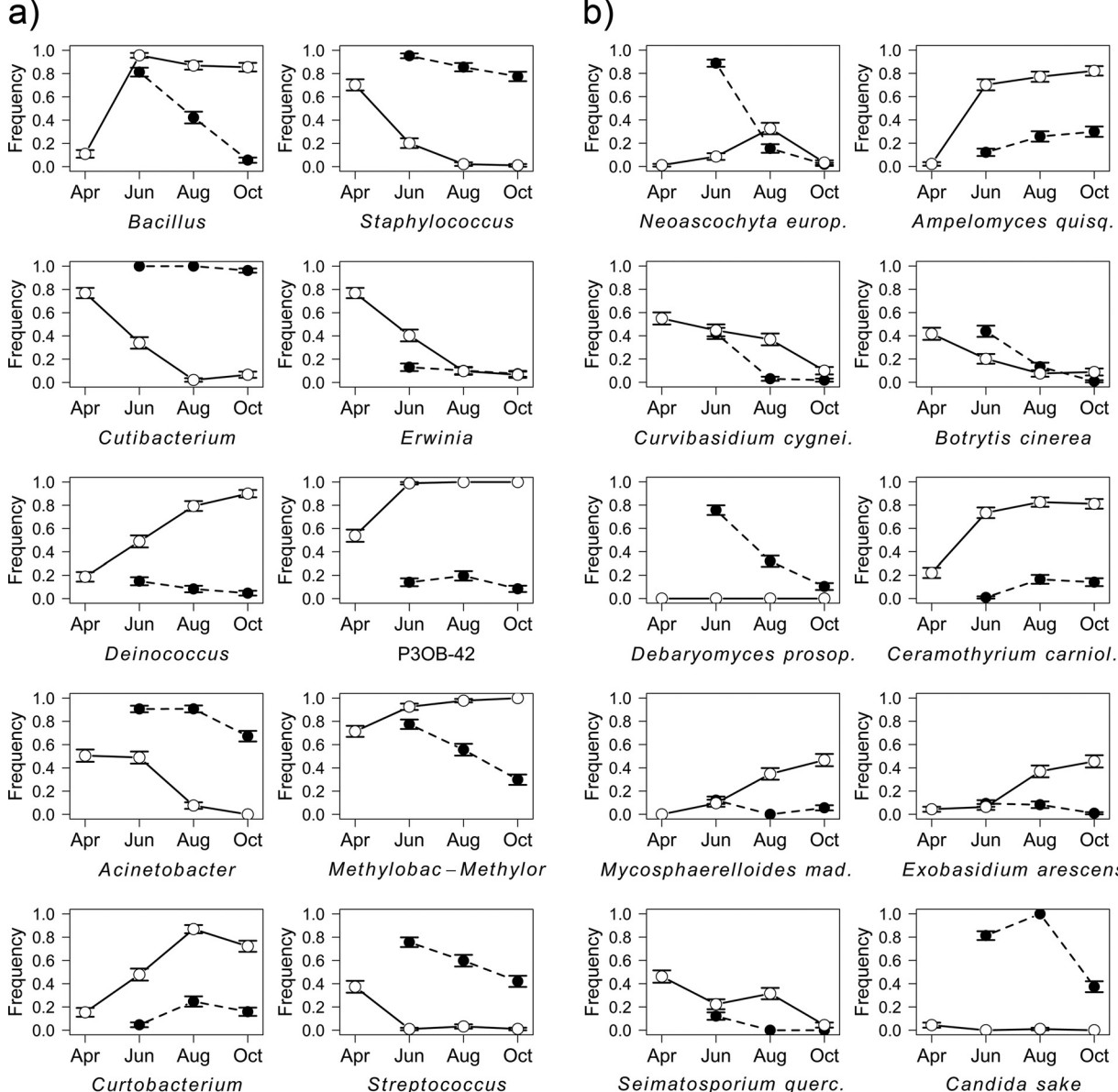

**FIG 5** Top 10 bacterial genera (a) and fungal species (b) with identified seasonal trends (mean ± SE) and their frequency in the samples collected at individual time points (April, June, August, and October). Empty circles and solid lines represent leaf samples, and full circles and dashed lines represent larval samples. Full taxa names and statistics derived from fitted GLMs with binomial distribution are listed in Table S6 in the supplemental material.

## DISCUSSION

In accordance with our hypotheses, we found pronounced shifts in leaf and larval microbiota as the season progressed. The composition of leaf microbiota changed with respect to both its components, and changes in fungal richness were more pronounced than those in bacterial richness. In larvae, bacteria and fungi showed consistent patterns; their composition changed, and richness decreased. The quantitative similarity between leaf and larval microbiota was relatively low for bacteria and decreased throughout the season, whereas fungal similarity increased and was relatively high. In both bacterial and fungal components, the $\beta$-diversity of larvae was higher than that of leaves. In both leaves and larvae, seasonality was, along with host taxonomy, the most important factor shaping microbial communities (see Table S5 in the supplemental material). Overall, our study confirmed the strong seasonal shifts in leaf bacterial or fungal microbiota found in evergreen (34–37) and deciduous trees (34, 39); we supplemented these results by

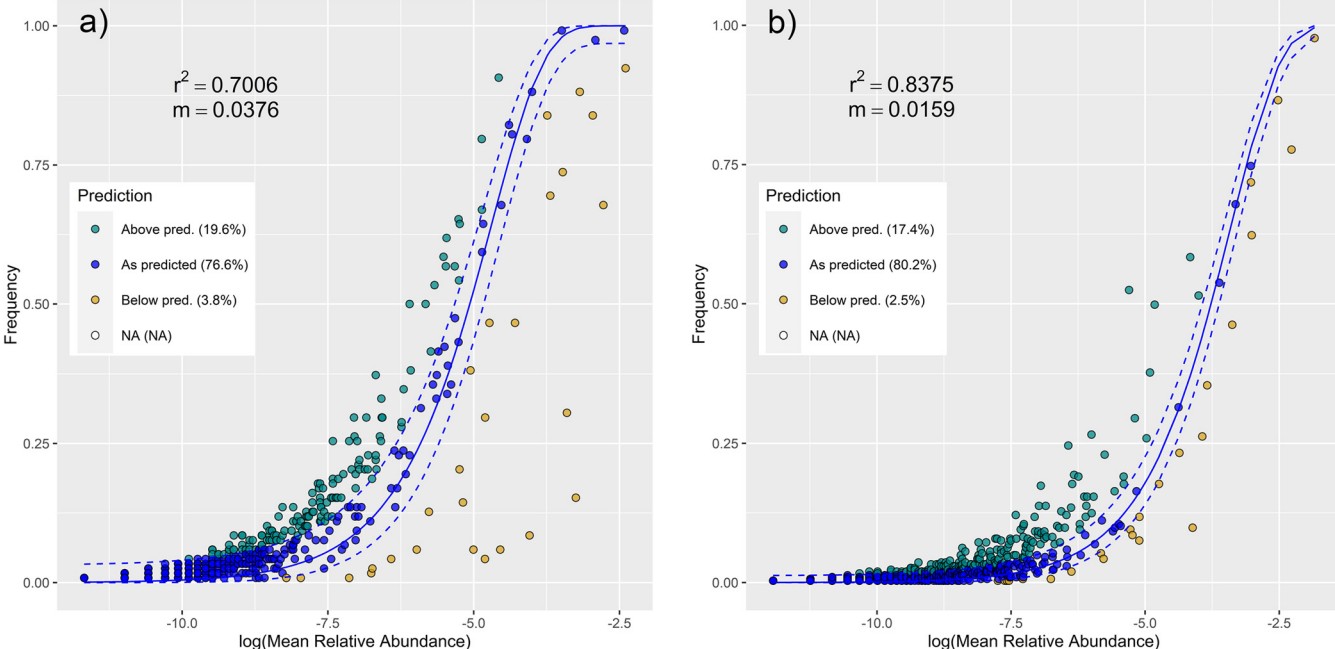

**FIG 6** Fit of the neutral community model (NCM) of larval community assembly for bacteria (a) and fungi (b) predicted occurrence frequencies for *x* and *y*. Solid blue lines indicate the best fit to the NCM according to reference 127, and the dashed blue lines represent 95% confidence intervals (CIs) around the model prediction. Bacterial genera/fungal species that occur more or less frequently than predicted by the NCM are shown in different colors. m, metacommunity size times immigration; $r^2$, the fit to this model.

simultaneously comparing bacteria and fungi and enhanced them by also describing the responses of microbiota of leaf consumers.

For both bacterial and fungal components of the leaves, the composition of the April samples was the most distinct from the other time points, especially with respect to the bacterial component. Young leaves differ fundamentally from older ones with regard to general leaf traits, such as nutrients and chemistry (41–43). Leaf traits (26, 47), environmental variables (i.e., climate), and changes in the local inoculum pool, combined with the host plant filtering, are fundamental drivers of leaf-associated communities (25, 34, 39, 48, 49), underlying the observed differences. Herbivory damage, which changes throughout the season, may also contribute to the seasonal shifts in leaf microbiota (50–52). Herbivores may potentially increase colonization by creating wounds and serving as vectors (53), thereby altering the size and diversity of microbial communities (51). Herbivory may also alter plant chemistry and nutrient supply (reviewed in reference 54) and stimulate or attenuate the plant's immune system, which can affect microbiota (55). The most abundant caterpillar taxa typically reach their peaks in spring, relatively soon after leaf emergence (56) and prior to their tanninization, which drives them to be less palatable (57). Therefore, we hypothesize that high herbivory rates in spring may be, at least partially, responsible for distinct leaf microbial assemblage.

In this study, the fungal richness of young leaves was high, and it sharply decreased in mature leaves (June), subsequently increased in the senescent leaves (August), and was the highest in leaves prior to abscission (October). Bacterial richness showed a similar trend; however, the changes were not significant. This shows that bacteria and fungi act similarly with respect to diversity changes during the season, although these trends are more pronounced in fungi. Generally, trends in leaf fungal richness complied with the successional model of community richness proposed by Jackson et al. (58). Fungi colonize leaves soon after or even prior to their emergence (59), reaching the highest richness under moderate temperatures and high moisture availability. Summer drought, high temperatures, UV radiation, and competition limit their occurrence (30, 60), reducing their richness (38). Subsequent resource diversity increases along with the occurrence of more favorable environmental conditions, leading to an

increase in richness in autumn (58). Our findings are in accordance with other studies that found high fungal diversity prior to leaf abscission (37, 38, 40).

Larval bacterial and fungal richness decreased throughout the season. In this study, most of the larval microbiota likely originated from the gut, as it is the largest absorbing part of the body; moreover, the most dominant intracellular symbionts (*Rickettsiales*) were excluded from the analyses. In larval Lepidoptera, the gut microbial diversity declines throughout the life cycle (10, 44) owing to changes in the physiological conditions of the gut lumen (61). However, the study species produce two or three generations per year (see Table S2 for references); this enabled us to sample similar instars throughout the season. Therefore, significant involvement of gut filtering is unlikely. Possibly, the decrease in richness resulted from changes in environmental conditions, which may alter microbial richness (62). Microbial community composition also changed throughout the season. In bacteria, however, the effect of larval species was more significant (17.64% of variability for larval species versus 4.66% for time point and 6.23% for the interaction of larval species and time point). In fungi, the seasonal shifts were more important than larval taxonomy, which was reflected only at the family level (6.82 versus 0.55%, respectively). Moreover, tree species had a significant effect on fungal composition (including an interaction with time point) that was more significant than its effect on bacterial composition. Collectively, our results comply with our previous study of free-feeding caterpillars that found a strong effect of larval species shaping gut bacteria but not fungi, where the effect of spatial variables was more important (50).

In lepidopteran larvae, gut microbiota often reflects that of their diet (9, 11–13). In this study, however, the quantitative similarity between larval and their respective leaf bacterial microbiota was only ∼9%. The similarity was higher for fungi (∼27%). To analyze leaf microbiota, we treated the leaf as a whole, that is, including both endophytes and epiphytes. However, to analyze the fungal similarity between leaves and larvae, we focused solely on endophytes, for which the presence in the inner tissue of leaf is very probable to obtain the best comparable minimum because leaf miners feed inside the leaf (i.e., on endophytes only), increasing the similarity to ∼53%. For bacteria, however, no tool for endophyte assignment was available; hence, endophyte selection was not possible. However, given that bacteria were analyzed at the genus level, and fungi at the species level, the actual leaf-larval bacterial similarity may be even lower. The overall low similarity of bacteria compared to fungi is in accordance with the results of our previous study on free-feeding caterpillars (50).

The leaf-larval similarity for bacteria decreased throughout the season, whereas fungal similarity increased. However, the factors causing this trend remain unclear. Generally, the different leaf-larval similarity of samples from different time points may have stemmed from (i) different leaf microbiota input (i.e., altered microbial composition of the diet), (ii) altered quality of the diet leading to favoring specific (functional) microbial groups, or (iii) different core microbiota input (i.e., life history-related patterns, whereby adult females sample a different microbial pool from the environment before oviposition, which is transmitted to the larvae).

Patterns of leaf-larval similarity throughout the season were evidenced by the analysis of the most frequent microbial taxa with significant seasonal trends. We identified those (i) frequent in larvae and less frequent/absent in leaves, (ii) frequent in leaves and less frequent in larvae, and (iii) frequent in both leaves and larvae. The first group comprised taxa with potentially beneficial effects for larvae, e.g., *Streptococcus* (a common insect gut inhabitant with proteolytic activity potentially helping the host to contend with plant secondary metabolites [63, 64]), *Candida sake* and *Debaryomyces prosopidis* (common colonizers of insect guts, playing a possible nutritional role [65, 66]), and *Neoascochyta europaea* (a dominant fungal associate of cricket guts [67]; synthesis of amino acids and oxidative processes in vertebrate guts [68]). Their levels were typically higher in July and decreased throughout the season. Given their low frequency in leaves (and even the absence in *C. sake* and *D. prosopidis*), they may have been either acquired via vertical transmission, whereby extracellular microbes are added to the egg surface by

the ovipositing female through the secretions or feces, which are then ingested by hatching larvae (10, 46, 61, 69), or ingested with diet in low concentrations (9) and subsequently multiplied in the larvae. Based on the neutral model, *C. sake* and *D. prosopidis* occurred in larvae more frequently than predicted. Therefore, we suggest that these species may represent potential larval core taxa.

Among the second group (more frequent in leaves), we identified those with a decreasing seasonal trend, *Erwinia* (common colonist of leaves [70]), *Seimatosporium quercinum* (colonist of oak twigs [71]), and *Curvibasidium cygneicollum* (endophytic phytopathogen or mycoparasite [72]). These taxa probably represent early colonists which are later suppressed due to community succession. Among the taxa with an increasing trend, we identified plant pathogens, e.g., *Curtobacterium*, *Ceramothyrium carniolicum*, and *Mycosphaerelloides madeirae* (73–75), as well as taxa potentially beneficial for the plant, e.g., bacterial strain P3OB-42 (*Myxococcaceae*, suppression of phytopathogens [76]), *Ampelomyces quisqualis* (suppression of pathogenic fungi [77]), and *Deinococcus* (biodegradation of organic pollutants [78]). We suggest that these taxa may have been increasingly recruited by the plants or only dominated as a result of community succession due to their high tolerance to UV and ozone stress (79–81). Neutral models revealed that *Curtobacterium*, *C. cygneicollum*, *C. carniolicum*, *M. madeirae*, and *A. quisqualis* occurred in larvae more frequently than predicted. All these taxa are predominantly plant pathogens but may also colonize various body parts of insects that may act as their vectors (82, 83). In the case of plant pathogens, vectoring by insects may eventually evolve into mutualism if the insects benefit from the infested plant (2). The mutualistic potential and functionality of these taxa thus remain to be investigated. Nevertheless, the taxa of the second group were generally much less frequent in larvae than in leaves, suggesting the significant involvement of host filtering (84).

Indeed, lepidopteran gut flora may be filtered via host diet by favoring microbes that digest compounds prevalent in food (85). This mechanism was evidenced by the third group (frequent in leaves and larvae) that consisted of taxa potentially beneficial for both plants and larvae, although it may also comprise taxa with occurrence optimum in both leaf and larval environments without providing any benefits. Among this group, we identified taxa with overall high frequencies in leaves and decreasing trends in larvae; e.g., *Bacillus* (in plants, antifungal activity [86], higher tolerance to UV radiation [87], plant growth-promoting, denitrifying, and organic pollutant-degrading function [34]; in larvae, proteolytic activity countering plant protease inhibitors [64]) and *Methylobacterium/Methylorubrum* (in plants, plant growth-promoting function, induction of systemic resistance against pathogens [88, 89]; in larvae, synthesis of amino acids and production of digestive enzymes and vitamins [90, 91]). Taxa prevalent in larvae and less frequent and decreasing in leaves were *Staphylococcus* (in plants, alleviation of biotic stresses [92]; in larvae, protease activity against plant secondary metabolites [SMs] [64]), *Acinetobacter* (in plants, potential denitrification and organic pollutant degradation [34]; in larvae, metabolizing of toxic phenolic glycosides [93]), and *Cutibacterium* (common gut core member of various arthropod taxa [94–96], possible digestive role [97]). The latter taxon occurred in larvae more frequently than predicted by neutral model, suggesting its potential functional importance in Lepidoptera. We also identified one "mixed" taxon, *Botrytis cinerea*, whose frequency was generally decreasing, being high in both leaves and larvae but was mostly higher in larvae. *B. cinerea* is a widespread plant pathogen (98) but also a mutualist of some Tortricidae, critically affecting their life cycle by synthesizing sterols (99). The ambivalent nature and potential importance of this fungus were evidenced also by neutral models, as its frequency was above prediction for both leaves and larvae.

Our results highlight the importance of considering seasonality in the studies of the interactions between plants, herbivorous insects, and their respective microbiomes. In this study, the endophagous larvae harbored species-specific bacterial (and less frequently, also fungal) communities consisting of core microbiome members. This supports the findings from our previous study on free-feeding lepidopterans (50). Observed patterns

suggest that microbes, especially bacteria, can play a more important role in the caterpillar holobiont than was generally assumed (9, 11). Many of these taxa may not originate from the diet, and the mechanism of their acquisition requires further investigation. Importantly, identification of biologically active taxa using transcriptomics is necessary to confirm their functional significance for the hosts (i.e., functional core; see reference 100 for terminology). In this study, we did not quantify microbial taxa, as only relative abundances were obtained. On the other hand, we used the Renkonen index for the comparisons of seasonal patterns, which is independent of abundance; moreover, in lepidopteran larvae, functionally important taxa may be present in disproportionally low abundances (101). Nevertheless, further studies using abundance-occupancy distributions combined with neutral models may help identify the drivers of the core microbiome assembly (102) and elucidate mechanisms responsible for the opposing seasonal trends in leaf-larval similarity of bacteria and fungi.

## MATERIALS AND METHODS

**Field sampling.** Sampling was conducted in 2-month intervals during a single growing season in 2018 in the temperate floodplain forests in Central Moravia, Czech Republic (PLA Litovelské Pomoraví; 49.6932°N, 17.1399°E). Leaves were sampled from April to October (four time points) and larvae from June to October (three time points), as they occur later in the season. We sampled the leaves of 5 tree species (Fagales) and 13 monophagous leaf-miner species (Lepidoptera: Gracillariidae, Tischeriidae) (see Table S1 in the supplemental material).

We set three remote sites (sampling plots) containing all tree species, each represented by six to seven individuals (Fig. S1, Supplemental File 1). For each tree individual, we measured the sampling height and diameter at breast height (DBH), both using a tape measure, and estimated the irradiated proportion of the crown. From each tree individual, we randomly selected five leaves, cut their middle portions (2 cm²; i.e., 10 cm² per sample) using sterilized tweezers and scissors, and transferred them to 1.5-mL tubes with 98% ethanol. Simultaneously, we manually sampled leaf miners; the whole leaves were placed into plastic containers and transferred to the laboratory where the larvae were extracted, their postmortem length was measured, and they were put into 1.5-mL tubes with 98% ethanol. To minimize the effect of the developmental stage on the composition and diversity of microbiota (10), we only selected larvae of similar lengths (instar) of the given species. Tubes with samples were stored at −32°C. In total, samples from 367 tree individuals and 311 larvae were selected for further processing (Table S1).

**Identification of larvae.** The identity of larvae was determined using field guides, online databases, and standard identification keys (Table S2). Also, 141 individuals (mainly congeneric species from the same host tree species) were determined through DNA barcoding of cytochrome oxidase subunit I (COI). We used DNA extracted from whole individuals (see below). COI barcodes were obtained using the general insect primers LepF1 and LepR1 (103). Each PCR (20 $\mu$L) consisted of 7.4 $\mu$L molecular biology-grade water (New England Biolabs, Ipswich, MA), 10 $\mu$L PCRBIO high-sensitivity (HS) Taq mix red polymerase (PCR Biosystems, London, England), 0.8 $\mu$M each primer (Sigma-Aldrich) and 1 $\mu$L template DNA. The amplification consisted of initial denaturation at 94°C for 3 min; 5 cycles at 94°C for 40 s, 45°C for 40 s, and 72°C for 1 min; 35 cycles at 94°C for 40 s, 51°C for 40 s, and 72°C for 1 min; and a final extension at 72°C for 5 min. PCR products were sequenced in the forward or reverse direction (Macrogen Europe, Amsterdam, Netherlands). Specimen records with sequences are accessible on BOLD (http://www.boldsystems.org/index.php/Public_SearchTerms?query=DS-MINS).

**Sample processing.** We prepared the surface-sterilized larvae and leaves containing both surficial and endophytic microbiota. Each larva was washed by vortexing in a 1.5-mL tube with 98% ethanol at 2,100 rpm for 90 s, transferred to a clean 1.5-mL tube, and washed with a 1-mL sterile solution of 1% Tween 80 and phosphate-buffered saline (PBS) solution (Sigma-Aldrich, Saint Louis, MO, USA) at 2,100 rpm for 45 s. The entire process was repeated (104), and each larva was transferred to a new 1.5-mL tube with 50 $\mu$L of 1× PBS. The leaf samples in 1.5-mL tubes were vortexed at 2,100 rpm for 45 s and centrifuged at 5,400 × $g$ for 15 min at 4°C. The supernatant was discarded, and the residual ethanol was evaporated at 55°C for 45 min. Subsequently, leaf samples were resuspended in 200 $\mu$L of 1× PBS. All samples were stored at −32°C for subsequent DNA isolation.

**DNA metabarcoding of bacteria and fungi.** DNA was extracted from individual larvae and leaf samples using a NucleoSpin tissue DNA isolation kit (Macherey-Nagel, Düren, Germany) following the manufacturer's protocol with a minor modification (initial denaturation at 95°C for 3 min; 35 cycles at 94°C for 30 s, 55°C for 60 s, and 72°C for 60 s; and a final extension at 72°C for 10 min). Since the dissection of the gut in microlepidoptera is problematic, the whole larvae were processed. Based on our previous experience (50), we used highly degenerate primers, which provide broad microbial diversity recovery from both leaves and guts and simultaneously reduce amplification of host plant DNA (chloroplasts). The fungal ITS2 rRNA gene region was amplified using ITS3_KYO2 and ITS4_KYO3 primers (105) and the bacterial V5-V6 16S rRNA region using 799F and 1115R primers (106, 107) with barcodes added to the 5′ end to enable sample identification. To minimize the stochastic amplification, we performed all PCRs in triplicate. The amplification of fungal and bacterial gene regions was performed using cycling profile and PCR composition as described in Sigut et al. (50). All PCR products were checked using 1.5% agarose gel.

Within each "plate library" (96 samples), triplicate PCRs of individual samples were pooled. From individual libraries, we excised the amplicons of specific lengths from the 2% agarose gel and purified them using a QIAquick gel extraction kit (Qiagen, Hilden, Germany). DNA concentration was measured using a Qubit double-stranded DNA (dsDNA) broad-range (BR) assay kit (Thermo Fisher Scientific), and concentrations were equalized to 20 ng/$\mu$L. We subjected individual plate libraries to DNA ligation of sequencing adapters and library-unique multiplex identifiers using Kapa hyper prep ki, and then quantified them using a Kapa library quantification kit (both Kapa Biosystems). We created one final library of bacterial samples and one of fungal samples at 7.5 ng/$\mu$L by pooling the equimolar proportions of individual plate libraries. The sequencing was performed at the Central European Institute of Technology (CEITEC; Masaryk University, Brno, Czech Republic) on NextSeq 500 for the bacterial library (one run, single end, $1 \times 150$-bp reads), and on MiSeq for the fungal library (four runs, paired end, $2 \times 300$-bp reads) (both Illumina Inc., San Diego, USA). Raw demultiplexed sequencing data with sample annotations are available at the NCBI BioProject website (https://www.ncbi.nlm.nih.gov/bioproject/) under the accession numbers PRJNA694554 for leaves and PRJNA814857 for leaf miners.

**DNA metabarcoding data processing.** Sequencing data were processed using QIIME 2.0 2020.2 (108). Raw reads were subjected to demultiplexing and quality filtering using the q2-demux plugin. In the case of the fungal data set, the ITS region was extracted using the q2-ITSxpress plugin (109). Subsequently, we used the DADA2 algorithm to denoise reads (110) and produced a feature table with counts of amplicon sequence variants (ASVs) per sample. To assign taxonomy, we used a trained naive Bayes classifier against the SILVA_138_SSURef_Nr99 bacterial reference database (111) and UNITE QIIME release for Fungi version 8.0 (112, 113) applied with the q2-feature-classifier classify-sklearn function (114). We produced an ASV table with 20,796,997 bacterial and 5,539,028 fungal reads. We identified contaminant ASVs using the decontam package (115) using the prevalence method with extraction controls as negatives (three per each 96-well plate), with the probability threshold for the rejection of noncontamination set to 0.1. We discarded 194 bacterial and 154 fungal ASVs (1.40% of reads [Table S3 - Supplemental File 2]) and removed reads associated with the chloroplasts and mitochondria (5.95%), archaea (16 reads) and those unassigned (2.41%). Finally, 18,426,696 bacterial (6,969,146 larval; 11,457,550 leaf) and 5,338,357 fungal reads (2,388,065 larval, 2,950,292 leaf) were used for analysis.

**Statistical analyses.** We analyzed the data in R 4.2.1 (116) and Canoco 5.01 (117). We used Krona charts (118) for hierarchical visualization of the recovered fungal and bacterial composition of the leaf and larval microbiota throughout the season. Bacterial ASVs were analyzed at the genus level (only a small number of ASVs could be classified to the species level), while fungal ASVs were analyzed at the species level. For the bacterial and fungal taxa, the number of reads, and the variables entering the analyzes, see Table S4 (Supplemental File 3). Most dominant *Rickettsiales* were excluded from the analyses of larvae, as they are intracellular parasites, most probably originating from the rest of the body instead of the gut lumen. However, they were not excluded from the analyses of leaves which they may colonize (119). Moreover, this group was excluded from the comparison of leaf and larval microbial composition (from both data sets). Fungal analyses were performed for all fungi, and then we analyzed separately only endophytic fungi, which were selected using the library FUNGuildR (120).

To determine the most important factors shaping the composition of the leaf and larval microbiomes, we performed a permutational multivariate analysis of variance (PERMANOVA) for the bacterial and fungal data sets separately, using the library vegan (121), with 999 permutations and distance matrices calculated using the Bray-Curtis method. For leaf microbiomes, we used two groups of explanatory variables, characterizing the (i) host tree individual (species, DBH, sampling height, irradiation of crown), and (ii) environment (sampling plot, time point). For larval microbiomes, a third group characterizing host larvae (species, family) was added. The final models were built by stepwise forward selection based on the Akaike information criterion (AIC), and potential interactions between explanatory variables were also checked. The resulting models were accompanied by partial redundancy analyses (p-RDA) and tested using the Monte-Carlo test with 999 permutations (with the exception of the fungal data set for larvae where a partial canonical correspondence analysis [p-CCA] was used owing to the long gradient of the response data, for which a unimodal method is suggested). The variable tree species was used in the multivariate analyses as a covariate when focusing on differences in composition among time points. The variable time point and tree species were used as covariates when focusing on differences in composition among larval species.

To compare bacterial genera/fungal species richness, the number of reads in each sample was rarefied to 1,000 reads for bacteria and 500 reads for fungi. We analyzed the rarefied richness using generalized linear models (GLMs) with gamma distribution (except fungal richness in larvae fitted using a linear model based on a data normality check using the Shapiro-Wilk normality test), built by stepwise forward selection based on AIC from an above-defined set of explanatory variables characterizing the tree, environment, and larva (for larval microbiomes).

To compare the leaf and larval microbiota, we calculated quantitative similarity between each larva and its host-tree leaf sample with the Renkonen index (122) using the proportion of reads attributable to each microbial genus/species to the total number of reads in the sample. We analyzed the effects of the above-defined explanatory variables on similarity using GLMs with gamma distribution (bacteria) and binomial distribution (fungi), both built by stepwise forward selection. We performed PERMANOVA to compare the bacterial and fungal composition between larvae and leaves, except for the first time point in leaves when the larvae were not sampled. As multivariate variation among the test groups may compromise PERMANOVA results in case of an unbalanced number of samples, we added the PERMDISP2 procedure for the analysis of multivariate homogeneity of group dispersions (variances) based on the Bray-Curtis distance, measuring the distance to group centroids (123). The models were

accompanied by CCA analyses for both data sets, and each model was tested using the Monte-Carlo test with 999 permutations.

To identify taxa that showed significant seasonal trends, we binarized the bacterial and fungal data sets to the presence or absence of data. For the analysis, we selected only fully identified bacterial genera/fungal species present in at least 100 samples. For data on the selected 52 bacterial and 39 fungal taxa, we individually built GLMs with binomial distribution and with (i) sample type (leaf × larva), and (ii) sample type plus seasonal trend and their interaction as explanatory variables. We compared the models using AIC and performed an analysis of deviance. Due to multiple comparisons, we adjusted $P$ values by false-discovery rate correction. For plotting, we selected 10 bacteria and 10 fungi with the highest differences in AIC and, thus, with the strongest simple seasonal trend or the strongest interaction of seasonal trend and sample type.

To quantify the involvement of neutral processes in larval microbial community assembly, we created neutral models using the libraries reltools (124), phyloseq (125), and GUniFrac (126). First, we rarefied samples to the same sequence depth, i.e., 1,000 reads for bacteria and 500 reads for fungi. Then, we fitted the neutral models (127) and extracted information about taxa fitting the null model or being above prediction or below prediction.

**Data availability.** Larval specimen records with sequences are accessible on BOLD (http://www.boldsystems.org/index.php/Public_SearchTerms?query=DS-MINS, data set DS-MINS). Bacterial and fungal raw demultiplexed sequencing data with sample annotations are available at the NCBI BioProject website (https://www.ncbi.nlm.nih.gov/bioproject/) under the accession numbers PRJNA694554 for leaves and PRJNA814857 for leaf miners. An overview of the bacterial and fungal taxa, the number of reads, and the variables entering the analyzes is included in Table S4 (Supplemental File 3).

## SUPPLEMENTAL MATERIAL

Supplemental material is available online only.
**SUPPLEMENTAL FILE 1**, PDF file, 0.9 MB.
**SUPPLEMENTAL FILE 2**, XLSX file, 0.04 MB.
**SUPPLEMENTAL FILE 3**, XLSX file, 4.6 MB.

## ACKNOWLEDGMENTS

We thank the members of the Laboratory of Insect Trophic Strategies for participation in field sampling and laboratory processing.

This study was supported by the Czech Science Foundation (GA22-29971S).

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
