## [Reviewer comments · Microbiology Spectrum]

Microbiology Spectrum

Seasonal shifts in bacterial and fungal microbiomes of leaves and associated leaf-mining larvae reveal persistence of core taxa regardless of diet

Hana Šigutová, Martin Šigut, Petr Pyszko, Martin Kostovcik, Miroslav Kolařík, and Pavel Drozd

Corresponding Author(s): Hana Šigutová, Ostravska univerzita

Review Timeline:

Submission Date:	August 15, 2022
Editorial Decision:	October 25, 2022
Revision Received:	November 23, 2022
Accepted:	December 16, 2022

Editor: John Chaston

Reviewer(s): Disclosure of reviewer identity is with reference to reviewer comments included in decision letter(s). The following individuals involved in review of your submission have agreed to reveal their identity: Yongqi Shao (Reviewer #1)

Transaction Report:

DOI: <https://doi.org/10.1128/spectrum.03160-22>

October 24, 2022

Dr. Hana Šigutová
University of Ostrava
Department of Biology and Ecology
Chittussiho 10
Ostrava 71000
Czech Republic

Re: Spectrum03160-22 (Seasonal shifts in bacterial and fungal microbiomes of leaves and associated leaf-mining larvae reveal persistence of core taxa regardless of diet)

Dear Dr. Hana Šigutová:

Thank you for submitting your manuscript to Microbiology Spectrum. I apologize for the delay, but one reviewer was unable to complete the review after initially accepting, and it added an unusual extension to the normal review time. Your manuscript has now been reviewed by two experts in the field, and you can see that both reviewers viewed your manuscript positively. Each raised some concerns, however, and I agree with these reviews. The reviewers are split on the importance of a qPCR analysis, and I would emphasize two lines of guidance: 1) if you choose to do the qPCR, it could be on a subset of the many samples you analyzed; 2) if you choose not to do the analysis, you should justify your reasoning sufficiently in both the text and reviewer response to address both reviewer's points, considering an aspect of Spectrum's scope that is relevant here: the work should be of high technical quality. I will likely defer to the reviewers' interpretations if the revisions fit this guideline.

If you choose to submit a revised version of your paper, please provide (1) point-by-point responses to the issues raised by the reviewers as file type "Response to Reviewers," not in your cover letter, and (2) a PDF file that indicates the changes from the original submission (by highlighting or underlining the changes) as file type "Marked Up Manuscript - For Review Only". Please use this link to submit your revised manuscript - we strongly recommend that you submit your paper within the next 60 days or reach out to me. Detailed instructions on submitting your revised paper are below.

Link Not Available

Sincerely,

John Chaston

Journals Department
Reviewer comments:

Reviewer #1 (Comments for the Author):

This study investigated seasonal shifts in the microbiomes of leaf mining larvae (Lepidoptera and Hymenoptera) and their host trees over one growing season in a deciduous temperate forest. The results not only highlight the importance of considering seasonality in the studies of the interactions between plants, herbivorous insects, and their respective microbiomes, but also clarified that microbes, especially bacteria, may play more important roles in the caterpillar holobiont than previously presumed. This is potentially an important contribution to the insect microbiome research field; however, there are several concerns that need to be addressed.

The authors collected both Lepidoptera and Hymenoptera insects, and since the biology of the two insect orders are too different, I would suggest to separate them in the analysis and then compare the two insect orders. Furthermore, is there any interspecies differences? Whether some insect hosts have more specific gut microbiota or not.

L343, "the changes were not significant", the statistical analyses (test method, P value...) should be clearly indicated in the related figure caption. Most of comparisons in the figures need be clarified.

High throughput sequencing is semi quantitative, representing only relative abundances. Validating some of the core taxa identified in this study (at least by Quantitative PCR) will provide more convincing evidence for the main conclusion here, because there is strong debate on the microbial residence of lepidopteran gut.

A table summarizing the taxa prevalent in larvae, in leaves and in both leaves and larvae as well as their potential functions (in the discussion) would provide more clear information.

Minors:

1. Table S5 Existing variables are less explanatory for microbial composition, and more variables should be introduced, such as temperature.
2. Table S5 Why not distinguish the sampling sites of insects, like trees (Sampling plot)? Even if the same insect, its microbiome may also vary depending on the sampling plot (see: Foliar-feeding insects acquire microbiomes from the soil rather than the host plant).
3. State the main results more clearly by adding relevant statistics to the graphs and legends. This is the basic requirement for figures and legends.
4. Fig. 8, Why choose frequency instead of showing the average relative abundance of the Top ten microorganisms?
5. L297-L299, relative abundances are useful information, microbiota composition of leaf and larvae throughout the season should be described.
6. L283-L289, please provide a complete table of statistical results.
7. L18, L89 "ITS2 rRNA metabarcoding", rRNA gene or rDNA like 16S rDNA, and why select this set of primer for fungi (ITS2 region)?
8. L392, wrong citations, the reference 84 studies *Enterococcus mundtii* but not *Streptococcus*, please check.

Reviewer #2 (Comments for the Author):

The authors describe the microbiota of insects and their leaf environments from a deep sample pool to address questions about the importance of the lepidopteran microbiota. The data are interesting given ongoing debate about the importance of the lepidopteran microbiota. In my opinion the data are beautifully and rigorously presented. The paper was a pleasure to read and the methods robust and informative.

However, I think there is an issue the authors should address to make their results clearer. The authors include a few Tenthredinidae in their analysis, but I think this might detract from the overall message. Does the message change if these few samples are excluded, so that the text's comparisons with other lepidopteran work are not confounded? If so, then this could drive the authors' detection of patterns that are at least somewhat in contrast to the work exemplified by Hammer. If not, I see no tangible benefit to retaining a few insect samples that otherwise convolute the analysis. I recommend the authors perform a re-analysis and drop the samples; or explain why they should be retained (I may be missing something important).

Some readers may wish to see quantitative estimates of bacterial abundance, as others have argued that this is an objective metric that can determine the 'true' microbiota. I'm not in this camp, but the authors may wish to proactively include some qPCR data. I think the data may not be necessary, because the authors comparisons, e.g. using the Renkonen index, are mostly independent of these concerns (the patterns emerge whether abundant or not). See also my comment about neutral models below, too, which also applies to patterns. But, readers will likely be interested in it and it would lend strength to the author's points.

I have several small suggestions:

L123 - form > from

Please provide brief methods details at L123 (Hrcek) and L130 (ref 48).

L135 - please describe the minor modification

L170 - were any archaeal reads detected and, if so, were they discarded?

L238 - please explain where the 12.8% etc values and the 7.87% etc values come from - are these a separate permanova comparing April to other time points, and then comparing within June, August, and October? Please clarify. Same question arises at L245-7

L288 - please edit this sentence for clarity (twice less?)

L320 - would 'relatively' be better than 'very'?

L326 - with > "by also describing"?

L332 - alters > changes

L362 - these numbers vary slightly from the table - please synchronize them.

L440 - the authors allude to neutral modeling; I think using the `sncm_fit` method of Sloan 2006 as implemented in Burns 2016 (<https://www.nature.com/articles/ismej2015142>) would be trivial, and could provide an additional layer of insight. I know burnout on additional experiments is real, but my lab recently tried it and it only takes a few minutes to get the script running. I'd be happy to provide some guidance or links if the authors are interested to try it.

Staff Comments:

Preparing Revision Guidelines

Please return the manuscript within 60 days; if you cannot complete the modification within this time period, please contact me. If you do not wish to modify the manuscript and prefer to submit it to another journal, please notify me of your decision immediately so that the manuscript may be formally withdrawn from consideration by Microbiology Spectrum.

23 November 2022

Dr John Chaston

Dear Dr Chaston,

Thank you for considering our manuscript entitled “**Seasonal shifts in bacterial and fungal microbiomes of leaves and associated leaf-mining larvae reveal persistence of core taxa regardless of diet**” (Spectrum03160-22) for publication. I, along with my co-authors, would like to re-submit its revised version.

We received two positive reviews on our manuscript. We carefully checked the manuscript and made appropriate changes in accordance with the reviewers’ suggestions. The most extensive change to emerge from the reviews was the omission of hymenopteran samples from the analyses. However, the results and the main significance remained unchanged. We also decided not to include qPCR analysis in our study (see the reasoning below). We believe that the inspiring and relevant comments of both reviewers have significantly improved the overall quality of our manuscript. Our responses are attached herewith.

We look forward to hearing from you regarding our submission. We would be glad to respond to any further questions and comments that you may have.

Sincerely,

Hana Šigutová

Department of Biology and Ecology
University of Ostrava, Czech Republic
hana.sigutova@osu.cz

Editor

Thank you for submitting your manuscript to Microbiology Spectrum. I apologize for the delay, but one reviewer was unable to complete the review after initially accepting, and it added an unusual extension to the normal review time. Your manuscript has now been reviewed by two experts in the field, and you can see that both reviewers viewed your manuscript positively. Each raised some concerns, however, and I agree with these reviews. The reviewers are split on the importance of a qPCR analysis, and I would emphasize two lines of guidance: 1) if you choose to do the qPCR, it could be on a subset of the many samples you analyzed; 2) if you choose not to do the analysis, you should justify your reasoning sufficiently in both the text and reviewer response to address both reviewer's points, considering an aspect of Spectrum's scope that is relevant here: the work should be of high technical quality. I will likely defer to the reviewers' interpretations if the revisions fit this guideline.

Thank you for your overall positive evaluation of our manuscript. Regarding the split opinions of the reviewers on the importance of qPCR, we would like to point out that the low abundance and limited function of lepidopteran gut microbiome has been questioned (e.g., Chen et al. 2019,

<https://doi.org/10.1002/ps.5642>); moreover, functionally important taxa may be present in disproportionately low abundances (Xia et al. 2017, <https://doi.org/10.3389/fmicb.2017.00663>). Based on your suggestion on how to deal with this aspect, we chose the second option (not to do the qPCR analysis) and addressed carefully all the reviewers' points, including addition of neutral models, as suggested by the second reviewer. We also added the brief justification to the discussion, as you suggested (lines 501–507). We believe that adding neutral models gave our manuscript an additional layer of insight regarding lepidopteran microbiome assembly (especially regarding the identification of potential “core” taxa), and that our manuscript is technically sound and robust, meeting all standards of *Microbiology Spectrum*. More detailed responses are given below.

Reviewer #1

This study investigated seasonal shifts in the microbiomes of leaf mining larvae (Lepidoptera and Hymenoptera) and their host trees over one growing season in a deciduous temperate forest. The results not only highlight the importance of considering seasonality in the studies of the interactions between plants, herbivorous insects, and their respective microbiomes, but also clarified that microbes, especially bacteria, may play more important roles in the caterpillar holobiont than previously presumed. This is potentially an important contribution to the insect microbiome research field; however, there are several concerns that need to be addressed.

The authors collected both Lepidoptera and Hymenoptera insects, and since the biology of the two insect orders are too different, I would suggest to separate them in the analysis and then compare the two insect orders. Furthermore, is there any interspecies differences? Whether some insect hosts have more specific gut microbiota or not.

Thank you for this comment; the other reviewer also pointed out that confounding Hymenoptera and Lepidoptera may distort the overall message of the paper. Based on your and his/her suggestion, we decided to completely omit Hymenoptera from the analyses (there were only 21 hymenopteran samples) to avoid confounding the results, and to be able to make more meaningful comparisons with other Lepidoptera literature. The main results remained unchanged (L 105–107, 258 and onwards). We also made appropriate changes in all supplementary figures and tables. Regarding potential interspecific differences (whether some insect hosts have more specific gut microbiota or not), there weren't any differences in leaf–larval similarity among species (bacteria: $df = 268$, $F = 1.07$, $P = 0.386$; fungi: $df = 270$, $\chi^2 = 5.29$, $P = 0.871$). We also analyzed differences in microbial composition among different larval species, and there were significant differences (in bacteria, host species was the variable explaining the highest portion of variability; L 282–283).

L343, "the changes were not significant", the statistical analyses (test method, P value...) should be clearly indicated in the related figure caption. Most of comparisons in the figures need be clarified.

Done (line 793 and onwards).

High throughput sequencing is semi quantitative, representing only relative abundances. Validating some of the core taxa identified in this study (at least by Quantitative PCR) will provide more convincing evidence for the main conclusion here, because there is strong debate on the microbial residence of lepidopteran gut.

Thank you for this comment. However, we do not think that including qPCR is necessary (and the other reviewer has the same opinion). First, we used Renkonen similarity index for the comparisons, which is independent of abundance (the patterns emerge regardless of abundance). Second, for the analysis of the

taxa with seasonal trends (based on which we identified potential “core” taxa), we chose those with higher frequency in the samples to avoid making overstated conclusions based on outliers (certain taxa may be very abundant but only in a few samples). Third, in lepidopteran guts, functionally important taxa may be present in disproportionately low abundances (Xia et al. 2017, <https://doi.org/10.3389/fmicb.2017.00663>), which means that abundant does not necessarily equal functionally or otherwise important (abundant taxa may be only first colonizers with competitive advantage via priority effect, or are strong competitors, have their optima there, etc.). Finally, based on the suggestion of the second reviewer, we added neutral models to quantify the involvement of neutral processes in microbial community assembly, and to make our hypotheses about core taxa more supported. More detailed answers are given below.

A table summarizing the taxa prevalent in larvae, in leaves and in both leaves and larvae as well as their potential functions (in the discussion) would provide more clear information.

Based on your suggestion, we added information about prevalence environment and potential function to Table S6. We also highlighted taxa that occurred more frequently than predicted by neutral models to make our hypotheses about core taxa more supported (in Table S6 and in the discussion, L 418–420, 430–435, 452–3, 456–8). Kindly note that there were taxa other than those highlighted in Table S6 that occurred more frequently than predicted; however, as our study aims primarily at seasonal differences, we decided to keep it uncluttered and focused on seasonal trends only.

Minors:

1. Table S5 Existing variables are less explanatory for microbial composition, and more variables should be introduced, such as temperature.

Thank you for this comment. In our analyses, we included % irradiation as explanatory variable for both leaf and larval microbiomes. Irradiation is known to strongly affect microbial communities and is tightly linked to moisture availability and temperature stability (Šigut et al. 2022, <https://doi.org/10.1038/s41598-022-19855-5>; Unterseher et al. 2007, <https://doi.org/10.1007/s11557-007-0541-1>; Gilbert et al. 2007, <https://doi.org/10.1890/05-1170>; Copeland et al. 2015, <https://doi.org/10.1094/MPMI-10-14-0331-FI>; Stone and Jackson 2020, <https://doi.org/10.1007/s00248-020-01564-z>). The average temperature also correlates with time point that was included as explanatory variable, too. While there was a significant effect of time point on both leaf and larval microbiomes, the variable % of irradiation explained only negligible portion of variability of bacterial composition of leaves (0.78%). We also mention the effect of temperature in the discussion (lines 362–367).

2. Table S5 Why not distinguish the sampling sites of insects, like trees (Sampling plot)? Even if the same insect, its microbiome may also vary depending on the sampling plot (see: Foliar-feeding insects acquire microbiomes from the soil rather than the host plant).

Sampling plot was tested as one of the explanatory variables in the analysis of larval microbiome, too. It is stated in the methods (lines 206–9). However, in Table S5, we present only variables that were significant for at least one of the components (bacteria or fungi) to keep the table uncluttered. To avoid doubts, we added this information to the caption of Table S5.

3. State the main results more clearly by adding relevant statistics to the graphs and legends. This is the basic requirement for figures and legends.

Done (L 793 and onwards).

4. Fig. 8, Why choose frequency instead of showing the average relative abundance of the Top ten microorganisms?

You probably meant Fig. 5. There were two main reasons we worked with frequency instead of relative abundances in that analysis. First, we worked with all species in parallel testing; an increase in the relative abundance of one taxon almost certainly leads to a decrease in the relative abundance of other taxa. In contrast, the presence/absence is relatively independent. Second, key or functionally important taxa may be present in relatively small abundances (Xia et al. 2017, <https://doi.org/10.3389/fmicb.2017.00663>), but what is important is their high frequency in the samples; after all, the neutral model is based on similar idea. For these two reasons, we decided to work with a relatively robust presence/absence data.

5. L297-L299, relative abundances are useful information, microbiota composition of leaf and larvae throughout the season should be described.

We modified the interactive Krona charts (Fig. S7 for bacteria and Fig S8 for fungi) to show the taxonomic composition of leaf and larval microbiota throughout the season.

6. L283-L289, please provide a complete table of statistical results.

Done (lines 301–319).

7. L18, L89 "ITS2 rRNA metabarcoding", rRNA gene or rDNA like 16S rDNA, and why select this set of primer for fungi (ITS2 region)?

Thank you for this comment, we corrected the term "ITS2 rRNA metabarcoding" to "ITS2 rDNA metabarcoding" (now lines 18 and 89). Regarding the choice of fungal primers, there are several options for amplification of various parts of the ITS and surrounding ribosomal coding regions. Based on the previous studies, the performance of metabarcoding based on ITS2 gene region may be comparable with other commonly used gene regions like ITS1 (Taylor et al., 2016, <https://doi.org/10.1128/AEM.02576-16>; Blaaid et al. 2013, <https://doi.org/10.1111/1755-0998.12065>). However, the consensus of the performance of various primer sets in recovering fungal diversity is missing and largely depending on the type of environmental sample. In our case, we aimed to amplify fungal DNA from the leaves and guts. Based on the literature, primers aiming at ITS2 region could provide better yield and diversity recovery in amplification from the samples containing large amount of non-target DNA. We selected primers ITS3_Kyo2 and ITS4_Kyo3 amplifying the ITS2 region (Toju et al., 2012, <https://doi.org/10.1371/journal.pone.0040863>) which, based on our previous experience (Šigut et al, 2022, <https://doi.org/10.1038/s41598-022-19855-5>), provide minimal amplification of host plant DNA and high fungal diversity recovery from both leaves and guts. We added this justification to the MS (lines 151–153).

8. L392, wrong citations, the reference 84 studies *Enterococcus mundtii* but not *Streptococcus*, please check.

Thank you for the notification, that was a mistake. We reformulated that sentence slightly and added relevant references (lines 408–10).

Reviewer #2

The authors describe the microbiota of insects and their leaf environments from a deep sample pool to address questions about the importance of the lepidopteran microbiota. The data are interesting given ongoing debate about the importance of the lepidopteran microbiota. In my opinion the data are beautifully and rigorously presented. The paper was a pleasure to read and the methods robust and informative.

However, I think there is an issue the authors should address to make their results clearer. The authors include a few Tenthredinidae in their analysis, but I think this might detract from the overall message. Does the message change if these few samples are excluded, so that the text's comparisons with other lepidopteran work are not confounded? If so, then this could drive the authors' detection of patterns that are at least somewhat in contrast to the work exemplified by Hammer. If not, I see no tangible benefit to retaining a few insect samples that otherwise convolute the analysis. I recommend the authors perform a re-analysis and drop the samples; or explain why they should be retained (I may be missing something important).

Thank you for this comment; the other reviewer had a similar idea. Based on your suggestion, we re-analyzed the data without the few hymenopteran samples and the main results remained unchanged. Therefore, we omitted Hymenoptera from the analyses (there were only 21 hymenopteran samples) to avoid confounding the results, and to be able to make more meaningful comparisons with other Lepidoptera literature (L 105–107, 258 and onwards). We also made appropriate changes in all supplementary figures and tables.

Some readers may wish to see quantitative estimates of bacterial abundance, as others have argued that this is an objective metric that can determine the 'true' microbiota. I'm not in this camp, but the authors may wish to proactively include some qPCR data. I think the data may not be necessary, because the authors comparisons, e.g. using the Renkonen index, are mostly independent of these concerns (the patterns emerge whether abundant or not). See also my comment about neutral models below, too, which also applies to patterns. But, readers will likely be interested in it and it would lend strength to the author's points.

Thank you for this comment; we agree that our comparisons are independent of abundance, and we believe that we can make meaningful conclusions based on the analyses as they currently stand. Nevertheless, based on your suggestion below, we added neutral models to add an additional layer of insight into the microbial community assembly of our samples (see below), especially regarding the identification of potential "core" taxa.

I have several small suggestions:

L123 - form > from

Thank you for the notification; however, this part has been deleted as we added methodological details based on your suggestion below.

Please provide brief methods details at L123 (Hreck) and L130 (ref 48).

Done (now lines 125–131; 137–143).

L135 - please describe the minor modification

Done (lines 149–150).

L170 - were any archaeal reads detected and, if so, were they discarded?

Yes, but there were only 16 archaeal reads detected (belonging to Methanobrevibacter and Methanosphaera). Regarding their low representation, they were analyzed along with bacteria in the former analysis. Nevertheless, we discarded those taxa in the reanalysis as there was no point in retaining them. We added this information to the MS (line 187).

L238 - please explain where the 12.8% etc values and the 7.87% etc values come from - are these a separate permanova comparing April to other time points, and then comparing within June, august, and October? Please clarify. Same question arises at L245-7

Thank you for this comment; clarified (lines 260–273).

L288 - please edit this sentence for clarity (twice less?)

Edited (lines 307–8).

L320 - would 'relatively' be better than 'very'?

Yes; edited (line 335).

L326 - with > "by also describing"?

You're right; that fits better. Edited (line 341).

L332 - alters > changes

Edited (line 349).

L362 - these numbers vary slightly from the table - please synchronize them.

Thank you for pointing out this discrepancy; corrected. Please note that omitting Hymenoptera from the analyses changed all those numbers slightly (lines 378–380).

L440 - the authors allude to neutral modeling; I think using the `sncm_fit` method of sloan 2006 as implemented in burns 2016 (<https://www.nature.com/articles/ismej2015142>) would be trivial, and could provide an additional layer of insight. I know burnout on additional experiments is real, but my lab recently tried it and in only takes a few minutes to get the script running. I'd be happy to provide some guidance or links if the authors are interested to try it.

That is a great idea; we agree that adding neutral modelling is a good way to provide additional information. We supplemented our analyses with neutral models (lines 245–49, 327–328), and our results were supported by those models. In the analysis of the taxa with significant seasonal trends, we also highlighted taxa that occurred more frequently than predicted by neutral model to make our hypotheses about core taxa more supported (in Table S6 and in the discussion, L 418–420, 430–435, 452–3, 456–8). Kindly note that there were taxa other than those highlighted in Table S6 that occurred in larvae more frequently than predicted by neutral models; however, as our study aims primarily at seasonal differences, we decided to keep it uncluttered and focused on seasonal trends only.

December 16, 2022

Dr. Hana Šigutová
Ostravska univerzita
Department of Biology and Ecology
Chittussiho 10
Ostrava 71000
Czech Republic

Re: Spectrum03160-22R1 (Seasonal shifts in bacterial and fungal microbiomes of leaves and associated leaf-mining larvae reveal persistence of core taxa regardless of diet)

Dear Dr. Hana Šigutová:

Your manuscript has been accepted, and I am forwarding it to the ASM Journals Department for publication. You will be notified when your proofs are ready to be viewed. Please note that your manuscript needs to add a Data Availability statement per Spectrum standards, and this can be added at a future stage in the process. You have already disclosed the accession numbers, it's just a matter of moving this to a new section. Thank you for considering Microbiology Spectrum for this story!

Sincerely,

John Chaston
Editor, Microbiology Spectrum

Journals Department
Supplemental Material: Accept
Supplemental Material: Accept
Supplemental Material: Accept